# OpenReview forum: "Testing the Limits of Jailbreaking Defenses with the Purple Problem"
_NeurIPS.cc/2024/Workshop/SafeGenAi — SafeGenAi Poster_

### Official Review · Reviewer_aXe4 · 2024-10-09
**Testing the Limits of Jailbreaking with the Purple Problem**

**Rating:** 7
**Confidence:** 2

**Review:**

The paper presents an interesting approach to investigating the jailbreaking defense pipeline for LLMs. Specifically, the authors define the Purple Problem, where the model cannot output the word purple.

Strengths:
- Clear problem setup of jailbreaking and inspection of the defense pipeline.
- The Purple Problem is an interesting distillation of real-world safety concerns, and the authors present thorough experiments for enforcement via fine-tuning weights and enforcement via preprocessing inputs.

---

### Official Review · Reviewer_Ucjv · 2024-10-09
**The paper lacks clear contributions and actionable insights compared to existing works**

**Rating:** 4
**Confidence:** 4

**Review:**

This paper presents an examination of current jailbreaking defenses in language models through a simplified test case, referred to as the "Purple Problem." It introduce a minimal definition of unsafe outputs (responses that contain the word "purple") to evaluate the robustness of defenses.

However, I am unclear on the specific contributions of this work. Previous research, such as [1], has already explored jailbreak defenses more comprehensively, considering a broader range of attacks and defenses, with similar findings that existing defenses cannot prevent all attacks. Additionally, the paper introduces the idea of dividing the defense pipeline into two stages: definition and enforcement, creating a dataset and evaluating defenses accordingly. Yet, from the experimental results, I do not see any actionable insights on how to strengthen defenses. If the paper could offer concrete suggestions based on the evaluation—such as how to improve defenses by focusing on these two stages—it would significantly enhance the paper's contribution.

[1] Zihao Xu, Yi Liu, Gelei Deng, Yuekang Li, and Stjepan Picek. 2024. A Comprehensive Study of Jailbreak Attack versus Defense for Large Language Models. In Findings of the Association for Computational Linguistics ACL 2024, pages 7432–7449, Bangkok, Thailand and virtual meeting. Association for Computational Linguistics.

---

### Official Review · Reviewer_ZfgN · 2024-10-09

**Rating:** 7
**Confidence:** 3

**Review:**

The paper distills the general problem of adversarial robustness on LLMs to a new problem called the Purple Problem, which defines an unsafe output as any output that contains the word "purple". The paper argues that if defenses cannot prevent models from outputting unsafe examples in this context, it casts doubt on whether they can be robust for more complicated unsafe outputs.

Strengths:

 - The paper is well-written, clearly explaining the motivation for the purple problem.
 - The paper experiments with many different kinds of attacks and defenses.

Weaknesses / Questions:
 - How is the purple problem affected by attacks such as PAIR or TAP, which modify the full prompt to obtain a jailbroken answer? The methods in the paper are either adversarial suffix attacks or preprocessing attacks (paraphrase aware attacks and high perplexity attacks). It would be interesting to see how the number of iterations from these kinds of attacks would change the Defense Success Rate.